# Radar Timing Range–Doppler Spectral Target Detection Based on Attention ConvLSTM in Traffic Scenes

**Fengde Jia** [1], **Jihong Tan** [1], **Xiaochen Lu** [1,*] **and Junhui Qian** [2]

[1] School of Information Science and Technology, Donghua University, Shanghai 201620, China; fdjia@dhu.edu.cn (F.J.); 2211849@mail.dhu.edu.cn (J.T.)
[2] School of Microelectronic and Communication Engineering, Chongqing University, Chongqing 400044, China; junhuiq@cqu.edu.cn
* Correspondence: lxchen09@dhu.edu.cn

**Abstract:** With the development of autonomous driving and the emergence of various intelligent traffic scenarios, object detection technology based on deep learning is more and more widely applied to real traffic scenarios. Commonly used detection devices include LiDAR and cameras. Since the implementation of traffic scene target detection technology requires mass production, the advantages of millimeter-wave radar have emerged, such as low cost and no interference from the external environment. The performance of LiDAR and cameras is greatly reduced due to their sensitivity to light, which affects target detection at night and in bad weather. However, millimeter-wave radar can overcome the influence of these harsh environments and has a great auxiliary effect on safe driving on the road. In this work, we propose a deep-learning-based object detection method considering the radar range–Doppler spectrum in traffic scenarios. The algorithm uses YOLOv8 as the basic architecture, makes full use of the time series characteristics of range–Doppler spectrum data in traffic scenarios, introduces the ConvLSTM network, and exerts the ability to process time series data. In order to improve the model's ability to detect small objects, an efficient and lightweight Efficient Channel Attention (ECA) module is introduced. Through extensive experiments, our model shows better performance on two publicly available radar datasets, CARRADA and RADDet, compared to other state-of-the-art methods. Compared with other mainstream methods that can only achieve 30–60% mAP performance when the IOU is 0.3, our model can achieve 74.51% and 75.62% on the RADDet and CARRADA datasets, respectively, and has better robustness and generalization ability.

**Keywords:** deep learning; millimeter-wave radar; radar range–Doppler spectrum; ConvLSTM; object detection

## 1. Introduction

With the continuous development of unmanned driving technology, the requirements for the safety and stability of vehicle driving are becoming more and more stringent. Advanced driver assistance technology (ADAS) plays a key role in this context. Various sensors are embedded in current intelligent driving vehicles, through which the vehicle perceives itself and its surrounding environment and makes judgments, such as cameras, laser radars, millimeter-wave radars, and other sensors. The two commonly used sensors are cameras and LiDARs. Cameras can capture a large amount of image information and provide rich semantic information. LiDARs can capture distance and speed information of surrounding objects. The information returned by these sensors can assist vehicles in better perceiving the surrounding environment and making timely adjustments according to changes in the environment, supporting the safe driving of the vehicle. Unfortunately, although the resolution of the information provided by cameras and LiDAR is very high, they still have limitations. The light sensitivity of cameras is very high, and they can perform well in dark or low-light conditions or under strong light during the day, but the semantic information that can be provided drops sharply. LiDAR performs poorly in bad

weather conditions such as rain, snow, and haze, posing a great safety hazard for vehicle driving. But there are some solutions. Due to poor angular resolution, large noise, and difficult-to-interpret signals, millimeter-wave radar has not been paid attention to in the past few years, but it can measure the speed and distance information of surrounding objects, send out electromagnetic wave signals, and is not affected by bad weather and light conditions, making up for the shortcomings of cameras and LiDAR and demonstrating better stability and reliability. Therefore, at present, millimeter-wave radar has been widely used in the military, aerospace, transportation, Internet of Vehicles, and other fields.

Object classification, target detection, and segmentation are the main tasks in the field of computer vision, among which detection and classification are important parts of the urgent development of intelligent driving. Facing complex traffic scenes, vehicles need to perceive the environment and respond quickly, which requires accuracy and real-time requirements. In the past ten years, deep learning has developed rapidly and made major breakthroughs. Related methods have been successfully applied to cameras and LiDARs [1–3]. However, due to the limitations of cameras and LiDARs, it is difficult to detect targets around the vehicle in harsh environments, so this paper refers to a new radar data format. Radar data can be represented as a list of targets (point cloud) or as a raw data tensor (range–Doppler or range–angle–Doppler map). The target list is the default radar data format and contains very low-level information such as the position, velocity, and radar cross-section [4] of targets around the vehicle. Point cloud data are usually processed by LiDAR, and information will be lost in harsh environments; the original data tensor is a relatively low-level and has a widely used data format, and will not lose a lot of information, so in this paper, radar range–Doppler (RD) maps are used as input for object detection. Since the use of range–Doppler data as the input data for traffic scene target detection is a new application scenario, there are few researchers working in this field, and there are few public radar tensor datasets. Fortunately, some foreign scholars have released related datasets, such as CARRADA, CRUW, and RADDet. This article uses the RADDET dataset [5] and the CARRADA dataset [6].

Target detection based on radar data is divided into two types; one is the traditional radar target detection method, based on constant false alarm rate (CFAR) detection and related variants, and the other is the currently more popular deep-learning-based target detection method. The constant false alarm detector is based on statistical theory [7] and judges whether the target exists through hypothesis testing. The specific implementation method is to set a threshold through the probability statistics method, set the sliding window to slide once on the range–Doppler spectrum, and compare the signal peak value of the sliding window with the previously set threshold. If the signal value is higher than the threshold, it is determined that there is a target at the current position; otherwise, it is considered that there is no target. CFAR detectors and their variants are usually an improvement in how the threshold is derived. Recently, deep-learning-based methods have become popular because of their good performance in high-dimensional and big-data processing. Using its achievements in image recognition and target detection [8,9], deep learning has been initially explored in radar target recognition and detection [10]. In this paper, we propose a new deep learning model for object detection and classification in complex traffic scenes via range–Doppler maps (RD maps). The model uses YOLOv8 as the basic skeleton, innovatively adds the ConvLSTM structure with a timing prediction function, and introduces the attention mechanism to capture key information to adapt to the complexity, timing of radar data, and small-scale features of radar objects, and improve the accuracy and real-time performance of detection as much as possible. Experiments on RADDet and CARRADA datasets show that our model can help improve object detection and classification performance on radar data and outperform other state-of-the-art detection methods. The contributions of this paper are mainly the following four points:

1. According to the characteristics of the dataset itself, due to the time–spatial relationship between the consecutive range–Doppler spectral frames in the dataset, we

innovatively designed a backbone network (ConvLSTM) with a time series prediction function to enhance the ability of the network to extract time series features;

2. For the characteristics of the range–Doppler spectrum, since the targets in the RD spectrum are very small, they are easily lost in the network detection process, so we introduce an improved lightweight and efficient channel attention mechanism (ECA) in the backbone and feature fusion parts, which improves the ability of the network to focus on key features;

3. The feature extraction network that combines timing prediction and attention enhancement shows good generalization ability in radar target detection tasks, and the detection performance on public datasets CARRADA and RADDet is better than other algorithms.

The organization of this paper is as follows. Section 2 introduces the basics of radar processing and some background work related to object detection. Section 3 then introduces our model. Experiments and results are collected in Section 4, while Section 5 discusses and concludes the paper.

## 2. Background and Related Work

### 2.1. Radar Sensor-Related Knowledge

Millimeter-wave radar is a radar system that uses millimeter waves for detection and imaging. Millimeter waves have strong penetrating power and high resolution, and can detect in environments such as bad weather and low illumination, so they are widely used. Radar target detection compares the emitted radar wave with the signal reflected by the target, and extracts the position, speed, size, and other information of the target from it for the subsequent detection process after processing.

One signal waveform commonly used in millimeter-wave radar sensors is frequency-modulated continuous wave (FMCW). The FMCW radar sends M chirp signals through the transmitting antenna, the sampling number of each chirp signal is N, and K receiving antennas will receive the return signal at the same time. The mixer will mix the received signal with the transmitted signal to obtain the intermediate frequency signal (IF), which produces an $N \times M \times K$ output tensor containing the received signal in the time domain. We refer to this tensor as an analog-to-digital conversion (ADC) signal. As shown in Figure 1, the distance information is extracted by performing a fast Fourier transform (FFT) on the ADC sample data in the distance dimension in a chirp, and then the distance Doppler spectrum (RD map) is obtained after an FFT in the velocity dimension. Finally, performing an FFT in the angle dimension to extract the angle information yields a range–angle–Doppler data tensor (or RAD cube). Because the amount of RAD cube data is too large, the subsequent calculation tasks are heavy, so this paper only uses the RD spectrum obtained by ADC data processing for target detection.

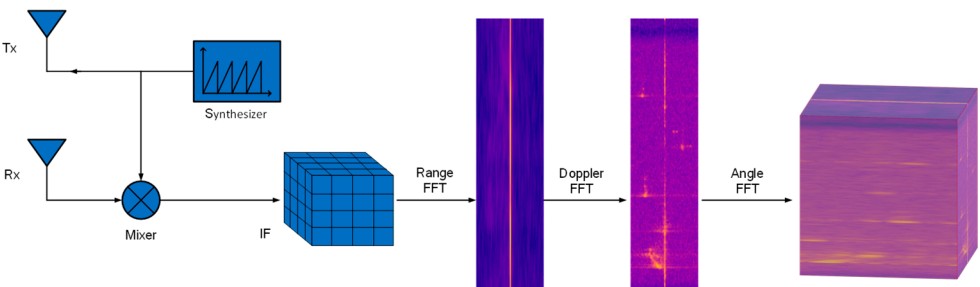

**Figure 1.** Schematic diagram of radar signal processing and RD spectrum generation (taking the CARRADA dataset as an example).

### 2.2. Traditional Approach Object Detection

For radar data, traditional target detection methods are mainly constant false alarm rate (CFAR) detection and improved variants. For a given clutter background, CFAR detec-

tion requires the evaluation of a suitable adaptive threshold, which is usually determined by estimating the clutter power of reference cells adjacent to the cell under test (CUT). However, the clutter statistics are unknown a priori and their estimation may be affected by the presence of interfering target echoes or the presence of sudden power transitions within the reference window. In this case, the detection performance of the radar system may suffer from a large decrease in the probability of detection (Pd) or a large increase in the probability of false alarm (Pfa). In fact, the presence of strong interfering targets will increase the detection threshold, thereby covering up weak primary targets, while the clutter edges will reduce the detection threshold, which will lead to an excessive false alarm rate when the CUT is in a high-power region. To overcome these situations and improve the detection performance, the most popular in the early stage is the Cell Average False Alarm Rate (CA-CFAR) detector [11,12]. It compares the energy of the detected unit, the CUT, with the threshold value obtained from the reference cell (RC) to make a judgment on whether the target exists or not. However, in multi-object detection, CA-CFAR tends to produce a masking effect [13], where weak objects may be ignored due to the large threshold caused by strong objects. Both the largest CFAR (GO-CFAR) [14] and smallest CFAR (SO-CFAR) [15] detectors are variants of CA-CFAR that split the reference window into spatial subsets before averaging. GO-CFAR is trained on the largest subset of clutter, while SO-CFAR uses the smallest. On the other hand, Variability Index CFAR (VI-CFAR) [16] adaptively selects a specific set of reference pixels to estimate background statistics. VI-CFAR is expected to degrade performance when the clutter distribution is complex and cannot be modeled by simple spatial partitioning. Ordered Statistical CFAR (OS-CFAR) [17] is an alternative to radar detection that obtains thresholds by sorting data samples and has better performance in cluttered edges and multi-object scenarios. Compared with CA-CFAR, OS-CFAR has the disadvantages of less detection rate loss with homogeneous clutter and a high computational cost. As a generalization of the OS-CFAR detector, Trimmed Mean CFAR (TM-CFAR) [18] estimates distribution parameters by averaging a set of ranked values. It should be noted that the above detection methods all correspond to one or more specific clutter distribution scenarios. However, realistic clutter is often difficult to predict, which often leads to poor detection performance of conventional CFAR detectors. Similar to the radar range–Doppler spectrum, the RD spectrum is susceptible to noise interference in the traditional detection process, which affects performance. Likewise, hyperspectral images suffer greatly from traditional processing methods. Therefore, some scholars use the mixed spatial attraction model (MSAM) based on linear Euclidean distance to obtain spatial correlation to better process images [19]. Other scholars proposed a new method called target-constrained interference minimization base station (TCIMBS), which can be used to select a subset of frequency bands for specific target detection while eliminating uninteresting targets and suppressing interference and background [20]. Some scholars have proposed an idea combining super sharpening and MS panchromatic sharpening technology for image fusion [21], which also provides a clear idea for us to explore new methods based on traditional methods.

### 2.3. Deep Learning Object Detection

With the continuous development of deep learning, target detection, as one of the tasks in the field of computer vision, is also undergoing iterative updates with each passing day. Currently, there are two mainstream algorithms for target detection: one is the one-stage detection algorithm represented by the YOLO algorithm, and the other is the two-stage detection algorithm represented by Faster-RCNN. Among them, the RCNN network proposed by Ross Girshick et al. is the originator of deep learning target detection papers. Considering that the generalization and robustness of features extracted by traditional image processing methods are not strong, convolutional neural networks are used to extract features [22]. Later, the improved Fast-RCNN and Faster-RCNN network models based on RCNN were successively introduced, but these models are all two-stage detection networks, mainly for identifying and positioning the region proposal. Although

the detection accuracy of this type of method is very high, because a separate network is required to extract the region proposal, it cannot break through the bottleneck in terms of speed, and it takes a lot of time to train and reason [23,24]. While the RCNN series was developing, Joseph Redmon and others proposed the first version of the You Only Look Once series for the first time, creating a precedent for one-stage detection algorithms. It regards the detection problem as a regression problem, using only one neural network to predict the location and category of the bounding box at the same time, so it is very fast, but the accuracy is not as good as the RCNN series [25]. Subsequently, a series of optimization strategy improvements were made for YOLOv1, and YOLOv2 [26] was launched. YOLOv3 adjusted the network structure for YOLOv2, using the Darknet-53 network structure, using multi-scale features for object detection and object classification with a logistic function instead of softmax, and achieved a good balance between detection speed and accuracy [27]. YOLOv4 uses Mosaic data enhancement, an anchor offset mechanism, a positive and negative sample matching strategy, and an improvement-of-loss function, and has achieved better detection results [28]. Compared with the previous version, the improvement of YOLOv5 mainly lies in the processing mechanism of the anchor, which can accelerate the convergence speed of the network model [29]. YOLOv6 is mainly used in industrial applications of target detection [30]. YOLOv7 tries to make the YOLO algorithm faster and better, while being able to support mobile GPU devices from the edge to the cloud [31].

These algorithms are based on a general framework for the detection and recognition of light-sensing images, and the pre-trained weights of the model are also suitable for most image datasets. Notably, while the image detection algorithm is developing, the corresponding datasets are gradually expanded, such as the ImageNet dataset [32], the COCO dataset [33], etc. These are target detections based on RGB images, which provide a data basis for the development of target detection.

Many researchers have also made efforts and contributions to apply deep-learning-based object detection algorithms to radar data of traffic scenes. Hsu Hao Wei et al. proposed a deep-learning-based convolutional neural network to reconstruct the RD map, so that the reconstructed RD map can be close to the RD map under FR [34]. Su et al. proposed a maritime target detection method based on radar signal graph data and graph convolution to apply graph-structured data to define detection units and represent temporal and spatial information of detection units [35]. Wang Chenxing et al. proposed a DL-based pulse Doppler radar UAV detection method [36]. He Jing et al. proposed a radar target detection method based on multi-task learning in a heterogeneous environment [37]. Wen Liwu proposed a two-step detection framework, in which feature differences and inter-frame correlations between moving objects and sea clutter are exploited for intra- and inter-frame detection, respectively [38]. Zheng Qiangwen proposed an object detection scheme based on distributed 1D CA-CFAR and region of interest (ROI) preprocessing [39]. RODNet proposed by Wang Yizhou et al. takes a small segment of an RF image as input to predict objects [40]. Wang Guohua et al. used the U-Net network for training and prediction [41]. Rodrigo Pérez et al. applied the real-time target detection system YOLO to preprocess the radar range–Doppler–angle power spectrum [42]. Roberto Franceschi et al. proposed a convolutional neural network-based model that can detect and localize targets in a multidimensional space of distance, velocity, azimuth, and elevation [43].

We were pleasantly surprised to find that Colin Decourt et al. proposed an end-to-end trainable architecture of hybrid convolution and ConvLSTM to learn the spatiotemporal dependencies between consecutive frames [44]. Lin Zhihui et al. proposed a novel self-attentional memory (SAM) to memorize features with long-range dependencies in both spatial and temporal domains [45]. Song Hongmei et al. further enhanced DB-ConvLSTM [46] with a PDC-like structure by employing several extended DB-ConvLSTMs to extract multi-scale spatiotemporal information. Zahidul Islam et al. proposed an efficient two-stream deep learning architecture LSTM (SepConvLSTM) using separable convolution and a pre-trained MobileNet network [47]. This provides us with effective ideas.

In the traffic scene, the radar data format such as the range–Doppler spectrum (RD maps) is obtained frame by frame and step by step, which has a certain timing. Therefore, this paper introduces a timing prediction function, the ConvLSTM model, into the YOLOv8 framework, so that the timing performance of radar data plays an auxiliary role in the network training and learning process. In addition, the attention mechanism is introduced to extract the features that the network considers to be important parts to improve the accuracy of detection.

## 3. Object Detection Method Based on Radar Range–Doppler Spectrum

The model we propose is an improved end-to-end trainable neural network based on the YOLOv8 model, which is mainly aimed at the improvement of the backbone network and the optimization of other structures. First, due to the traffic scene, the radar range–Doppler dataset has time series characteristics. Each frame in the dataset has a time–spatial connection with the preceding and preceding frames, and our network can easily capture and identify targets that are invisible to the naked eye through the temporal memory between the preceding and preceding frames. Therefore, for the backbone network, we innovatively introduced the ConvLSTM network with timing prediction capability and simplified its original structure so that it can be applied to the output of the upper-level CSP module. It also adapts its output to the input of the subsequent network. Second, due to the small and hidden characteristics of the target in the range–Doppler spectrum, it is difficult for the general target detection network to adapt to the detection of small targets in the RD spectrum. Each target only occupies a few pixels in the RD spectrum, and small targets such as pedestrians are even close to the point target, which is difficult for human eyes to distinguish. Therefore, by improving the lightweight and efficient ECA attention module, it is used to enhance the network feature extraction ability and the ability to focus on key features. We place it behind each CSP module in the backbone network and after the CSP module in the feature fusion area to enhance the feature extraction ability for small targets. In this section, we first preprocess the radar data through technical means so that it can match the input format requirements of our model, then introduce the model improvement method in detail, and finally introduce the experimental equipment for network training and inference.

### 3.1. Data Preprocessing

Before starting a new project, the first course of action is to process the original data to make it meet the input format requirements of the project. In this paper, we first perform dimensionality reduction processing on the 3D ADC cube complex data of the original dataset by summing the diagonal dimension tensors, and we only extract the information of the distance and Doppler dimensions. In order to save computing resources and speed up network training, we process complex data into real data, and finally obtain visualized RD spectra in batches. Since the tags in the original data are three-dimensional tags, after the radar data are dimensionally reduced, the tags of the corresponding objects are also processed into two-dimensional coordinate information. We organize the processed RD graph and corresponding labels into a new dataset format as input, and then send it to the network for training.

### 3.2. Object Detection Network Model

In this section, we propose a network architecture for object detection in range–Doppler spectra. Given the processed RD map as an input, a YOLOv8 network is used as the basic framework to design a network with time-series feature enhancement learning in the backbone. After an improved feature fusion layer, three YOLO heads are used for output. Based on the original YOLO v8, our model improves the detection performance in small target and complex background environments. The network structure diagram we designed is shown in Figure 2.

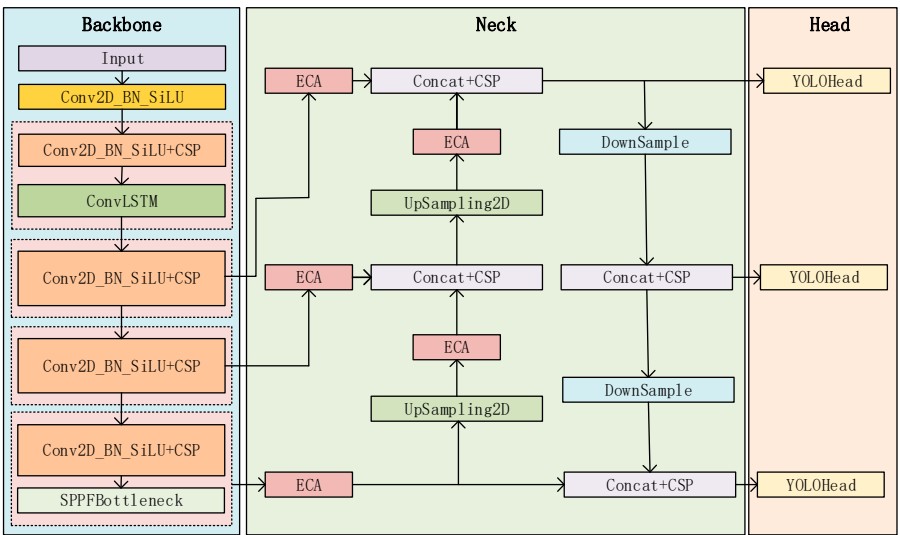

**Figure 2.** Object detection model architecture for range–Doppler maps.

### 3.2.1. YOLOv8 Algorithm Description

The model structure of YOLOv8 can be divided into three parts, namely the spine, neck, and head. Its network flowchart is shown in Figure 3. When the image is input, the backbone network mainly performs feature extraction. First, it performs a 2D convolution and then performs a 2D convolution to connect it with the CSP module through the residual structure, repeats this operation four times, and then achieves an adaptive size output through the SPPF module. The activation functions used here are all SiLU. SiLU has no upper bound, only a non-monotonic lower bound. It still maintains good performance on deep networks, which is conducive to improving the fitting effect of the model by increasing the depth. In order to fuse feature information of different scales, the neck will use three feature maps of different sizes extracted by the backbone for feature fusion. The deep feature layer is sequentially up-sampled and spliced to obtain a new feature layer, and then the shallow feature layer is sequentially down-sampled for the second splicing to obtain the final feature fusion layer. This can fully combine shallow features and deep features, make full use of semantic information, and speed up the efficiency of information dissemination. The head of the model uses the current mainstream decoupling head structure, which separates the classification and detection heads, and replaces Anchor-Based with Anchor-Free. In terms of Loss calculation, the TaskAlignedAssigner positive and negative sample matching strategy is adopted, and positive samples are selected according to the weighted scores of classification and regression scores. The Loss calculation part includes two branches: the classification and regression branches. There is no previous objectness branch. The classification branch still uses BCE Loss, and the regression branch needs to be bound to the integral representation proposed in Distribution Focal Loss, so Distribution Focal is used. Loss also uses CIoU Loss. The three Loss calculations are weighted with a certain weight ratio.

### 3.2.2. ConvLSTM

ConvLSTM is a neural network architecture combining a nonlinear convolutional neural network (CNN) and a long short-term memory network (LSTM), which is designed to process data with spatiotemporal information, such as video analysis or time series signals. Its basic idea is to replace the memory module of LSTM with external operations, taking advantage of CNN's advantages in processing spatial data such as images and videos. Its structure consists of an outer layer, an LSTM layer, and an output layer. At each time step, the input data are a three-dimensional vector (including the number of samples, width, and height), which is processed through the outer layers and then passed to the LSTM layer. Each unit in the LSTM layer contains a memory unit and three gating

units—input gate, forget gate, and output gate—and these gating units can control which information should be memorized, forgotten, and output in order to better process the time series data. Finally, the output layer converts the output of the LSTM layer into the desired output form. The basic mathematical model of ConvLSTM can be expressed as follows:

$$i_t = \sigma(W_{xi} * X_t + W_{hi} * H_{t-1} + W_{ci} \circ C_{t-1} + b_i) \tag{1}$$

$$f_t = \sigma\left(W_{xf} * X_t + W_{hf} * H_{t-1} + W_{cf} \circ C_{t-1} + b_f\right) \tag{2}$$

$$C_t = f_t \circ C_{t-1} + i_t \circ \tanh(W_{xc} * X_t + W_{hc} * H_{t-1} + b_c) \tag{3}$$

$$o_t = \sigma(W_{xo} * X_t + W_{ho} * H_{t-1} + W_{co} \circ C_t + b_o) \tag{4}$$

$$H_t = o_t \circ \tanh(C_t) \tag{5}$$

The structure diagram of ConvLSTM is shown in Figure 4:

Among them, $i_t$, $f_t$, and $o_t$ represent the input gate, forget gate, and output gate, respectively; $X_t$, $H_{t-1}$, and $C_{t-1}$ represent the input of the current cell, the output, and the state of the last cell, respectively; W and b represent the convolution kernel and bias; and σ is a function that assumes the shape of s. ConvLSTM changes the multiplication in the original LSTM to a convolution operation.

Since ConvLSTM can be used for video data captured by cameras, the datasets produced by radar RD spectra in real traffic scenes have continuity in time and space. Therefore, ConvLSTM can model the temporal dependence in continuous RD data and capture spatial information through convolution operations. This paper improves the original backbone framework of YOLOv8, introduces the ConvLSTM structure before the shallow feature fusion output layer, strengthens the feature extraction ability of the backbone network for RD time series data, and introduces the memory function. It should be noted that before the introduction of ConvLSTM, the output of the network needs to be converted into the data dimension that ConvLSTM allows for input. After passing through ConvLSTM, the output of the network needs to be converted into a data dimension that CNN can accept. This experiment shows that ConvLSTM is very effective for modeling time series data.

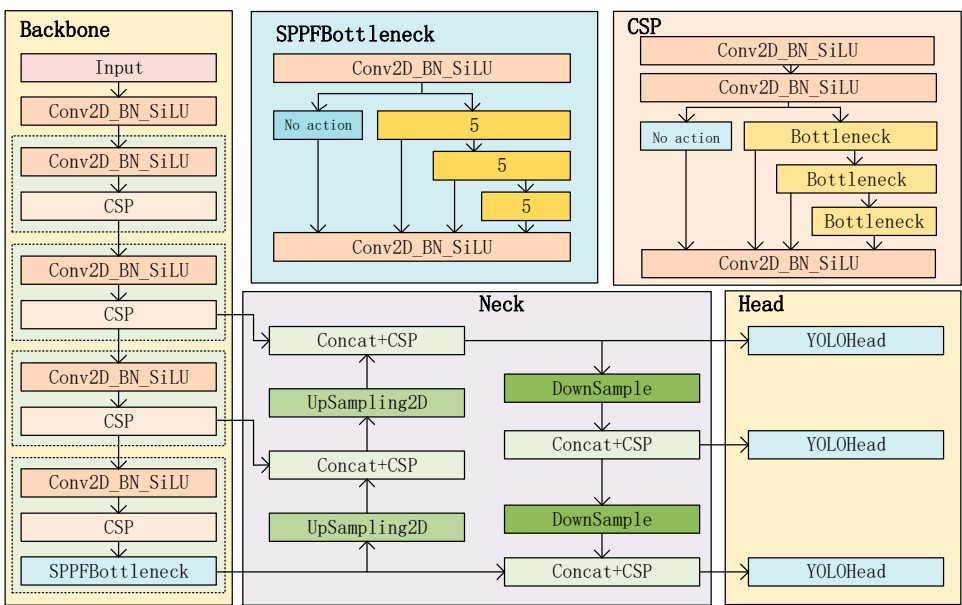

**Figure 3.** The original YOLOv8 model.

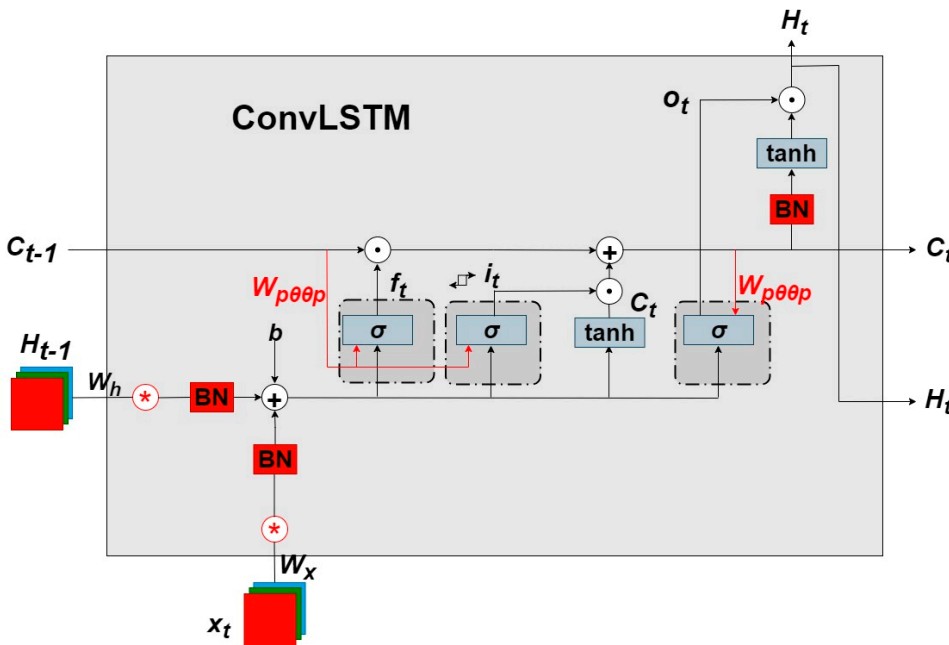

**Figure 4.** Schematic diagram of ConvLSTM structure.

### 3.2.3. Add Efficient Channel Attention (ECA) Module

The attention mechanism originated from the study of human vision, and the purpose is to make the model selectively focus on specific parts. In a recent study, a lightweight channel attention mechanism ECANet [48] was proposed, which obtained a local cross-channel interaction without dimensionality reduction under the premise of increasing the model complexity as little as possible, and it greatly improved the model performance. The model structure diagram of ECA is shown in Figure 5. It mainly improves the channel attention of SENet, removes the fully connected layer in SENet, replaces it with a $1 \times 1$ convolution kernel for feature processing, and determines the cross-channel interaction through the convolution kernel size k of a 1D convolution scope of information. This not only reduces the model parameters, but also makes the model lighter.

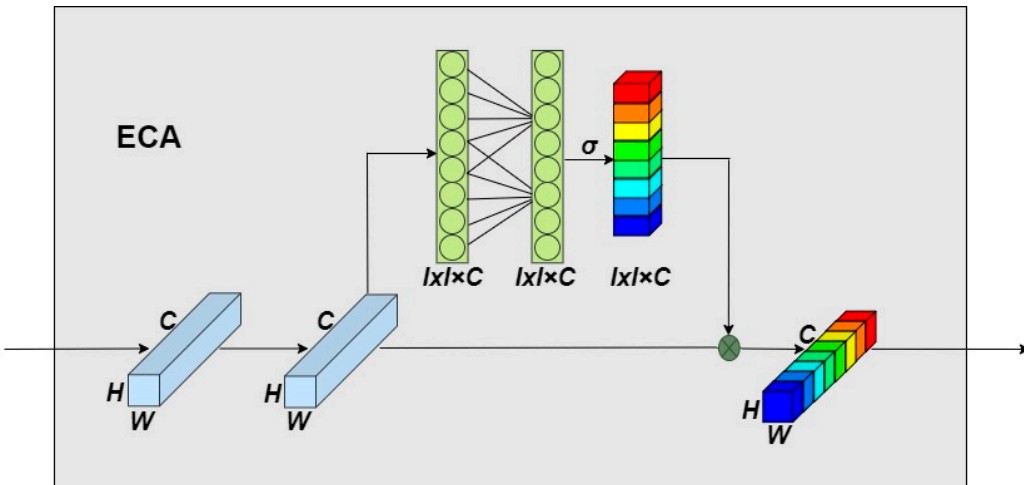

**Figure 5.** ECA attention mechanism structure diagram.

There are three feature output layers before FPN feature fusion and after up-sampling during feature fusion. The ECA attention mechanism is added to strengthen the network's focus on channels. This processing method can help the network emphasize important features while suppressing irrelevant features during feature fusion and splicing. This

improvement is beneficial to suppress the noise interference caused by the complex background in the RD image.

## 4. Experiments

The proposed model approach has been validated on real radar datasets of RADDet and CARRADA traffic scenes. For the completeness of experiments, and also to evaluate our model, in this section, we contrast and evaluate the proposed method with current popular object detection methods. The comparison methods include the two-stage detection algorithm Faster-RCNN, and the one-stage detection algorithms YOLOv3, YOLOv5, YOLOv7, YOLOv7-tiny, and YOLOv8.

### 4.1. Experimental Data

Our experimental data come from the public dataset RADDet provided by researchers at the University of Ottawa in Canada and the CRRADA dataset provided by ArthurOuaknine et al. from McGill University. Among them, the RADDet dataset provides six traffic scene categories, namely person, bicycle, car, truck, motorcycle, and bus. The CARRADA dataset provides three categories, namely pedestrian, cyclist, and car. The corresponding numbers of specific categories can be seen in Table 1. The original RADDet and CARRADA datasets provide 3D radar complex data (ADC cube) with dimensions of $256 \times 256 \times 64$, which will cause a serious computational burden during network training. Therefore, we preprocess the ADC data into two-dimensional real data. Figure 6 shows an RD spectrum in the RADDet dataset and the targets in it. Since there is a timing prediction module, ConvLSTM, in the network, it requires data input in the form of samples, one of which contains 10 frames of RD images, so the dataset should be delivered in the form of multiple samples. In this way, the learning ability of the network for time series prediction feature extraction can be maximized. We randomly divided the data into training set and testing set according to the ratio of 9:1, and randomly selected 10% of the training set as the validation set. In order to increase the reliability of the experimental results, we re-divided the distribution ratio of the dataset, randomly divided the original data into the training set and the test machine according to the ratio of 7:3, and randomly selected 10% of the training set as the verification set. The specific number can be seen in Table 1.

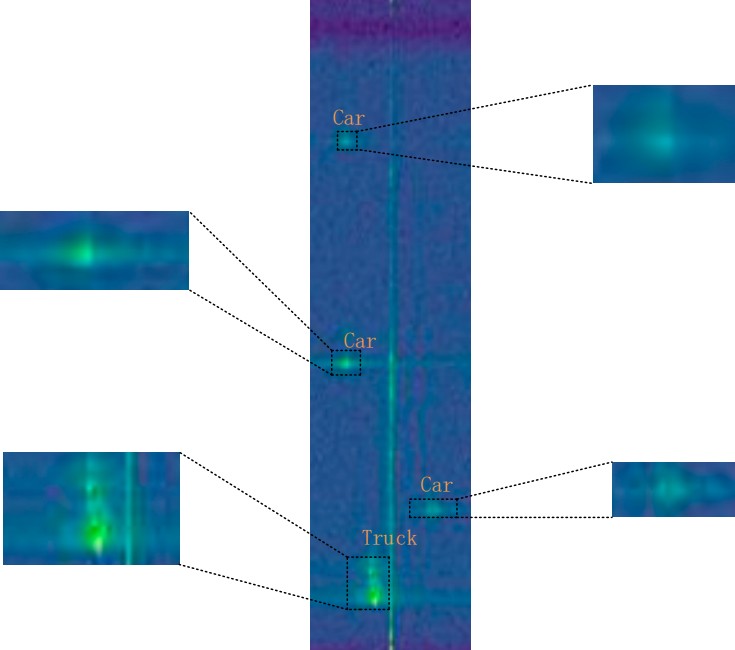

**Figure 6.** RD diagram representation of the RADDet dataset, with bounding boxes around objects and their scaling.

**Table 1.** Introduction to the number of categories of the two datasets and the division of the training set, validation set, and test set.

| Dataset | Categories | Quantity | Train | 9:1 Validation | Test | Train | 7:3 Validation | Test |
|---|---|---|---|---|---|---|---|---|
| RADDet | person | 4707 | | | | | | |
| | bicycle | 654 | | | | | | |
| | car | 12,179 | | | | | | |
| | truck | 2764 | 8227 | 915 | 1016 | 6399 | 711 | 3048 |
| | motorcycle | 56 | | | | | | |
| | bus | 154 | | | | | | |
| CARRADA | pedestrian | 2908 | | | | | | |
| | cyclist | 1595 | 5826 | 647 | 720 | 4531 | 504 | 2158 |
| | car | 3375 | | | | | | |

*4.2. Evaluation Indicators*

We evaluate our models using mean precision (mAP), a well-known metric for evaluating object detectors, providing both precision and recall at intersection-over-union (IOU) thresholds of 0.3 and 0.5. Their specific calculation methods are as follows:

- Accuracy: Refers to the probability of detecting the correct value among all detected targets:

$$\text{Precision} = \frac{TP}{TP + FP} \tag{6}$$

- Recall rate: Refers to the probability of correct identification in all positive samples:

$$\text{Recall} = \frac{TP}{TP + FN} \tag{7}$$

- AP: Refers to the average value of the detector in each Recall case, corresponding to the area under the PR curve:

$$\text{AP} = \int_0^1 P(r)dr \tag{8}$$

- mAP: The average evaluation of AP from the category dimension, so the performance of multi-classifiers can be evaluated:

$$\text{mAP} = \frac{1}{C} \sum_j^c AP_j \tag{9}$$

*4.3. Comparison Method and Training Details*

In order to evaluate our proposed model, we choose other current mainstream object detection methods for comparison. The comparison methods mainly include Faster RCNN, YOLOv5, YOLOv7, YOLOv7-tiny, YOLOv8, and the RADDet method and DAROD method proposed by other researchers. The data of all models adopt the RD graph provided by the preprocessed RADDet and CARRADA datasets and use the pre-trained weights provided by the model for training.

As shown in Tables 2 and 3, we provide settings about the experimental environment as well as model parameters. For all subsequent experiments, unless otherwise specified, the parameters shown in the table shall prevail. All models are trained for 300 epochs. In order to speed up the training, we freeze the training of the backbone network part in the first 50 rounds, and set the batch size to 4. We use the Adam optimizer, the β coefficient is set to 0.9, the weight decay is $5 \times 10^{-4}$ (we train both models with Adam optimizer with betas 0.9, 0.999, weight decay $5 \times 10^{-4}$), the maximum learning rate of the model is set to $1 \times 10^{-3}$, the minimum learning rate is set to $1 \times 10^{-5}$, and the learning rate changes according to the mathematical law of cosines. This experimental setup applies

to all comparison methods in this paper. All experiments use the Pytorch deep learning framework and NVIDIA GeForce RTX 2080Ti GPU for network training and testing.

**Table 2.** Experimental environment.

| Environment | Versions or Model Number |
|---|---|
| CPU | i7-1165G7 |
| GPU | RTX 2080Ti |
| OS | Windows 10 |
| Python | 3.6.13 |
| Pytorch | 1.10.2 |
| Torchvision | 0.11.3 |
| OpenCV-Python | 4.1.2.30 |

**Table 3.** Experimental parameters.

| Input Size | Optimizer | Momentum | Batch Size | Epoch | Learning Rate | Training and Test Set Radio |
|---|---|---|---|---|---|---|
| 640 × 640 | SGD | 0.937 | 4 | 300 | $1 \times 10^{-3}$ | 9:1 |
| 640 × 640 | SGD | 0.937 | 4 | 300 | $1 \times 10^{-3}$ | 7:3 |

*4.4. Analysis of Results*

Figure 7 shows the comparison results of mainstream methods in the field of target detection on the RADDet dataset. It is not difficult to find that, except for our model, other methods have missed detection, and the confidence of detecting objects is not high. One thing that stands out about YOLOv5 is that it can detect cars that other models cannot, except ours. Surprisingly, it misses the bus target and does not deliver a satisfactory result. Figure 8 shows the comparative results of various object detection methods on the CARRADA dataset. Due to the small number of categories in the CARRADA dataset, there are only one or two targets in each range–Doppler spectrum, and there are fewer occlusions between objects, so there are fewer missed detections. However, Faster RCNN still misses detection, which may be related to the detection performance of the algorithm. For the YOLO series of algorithms, objects can basically be detected, but there are differences in the detection confidence. At a glance, our model has the best detection performance.

Tables 4 and 5 show the performance of several different object detection methods on the RADDet and CARRADA datasets under the premise of dividing the dataset at a ratio of 9:1 and 7:3. In particular, since the time spent in each training epoch is similar during the model training process, the training time and testing time in Tables 4 and 5 are based on the time spent by each model in a training epoch. The best results are highlighted in bold, and the second-best results are underlined. For a more intuitive observation, Figure 9 shows the mAP of each model when the IOU is 0.3 and 0.5. It can be seen from the figure and table that among the seven target detection methods, the performance of our proposed model on the two datasets is significantly better than other methods in most cases, and it remains competitive with the YOLOv7 method on the RADDet dataset. The CARRADA dataset maintains competition with Faster RCNN, but there is also a big gap. The above results are all obtained by taking the best weight for detection after the training.

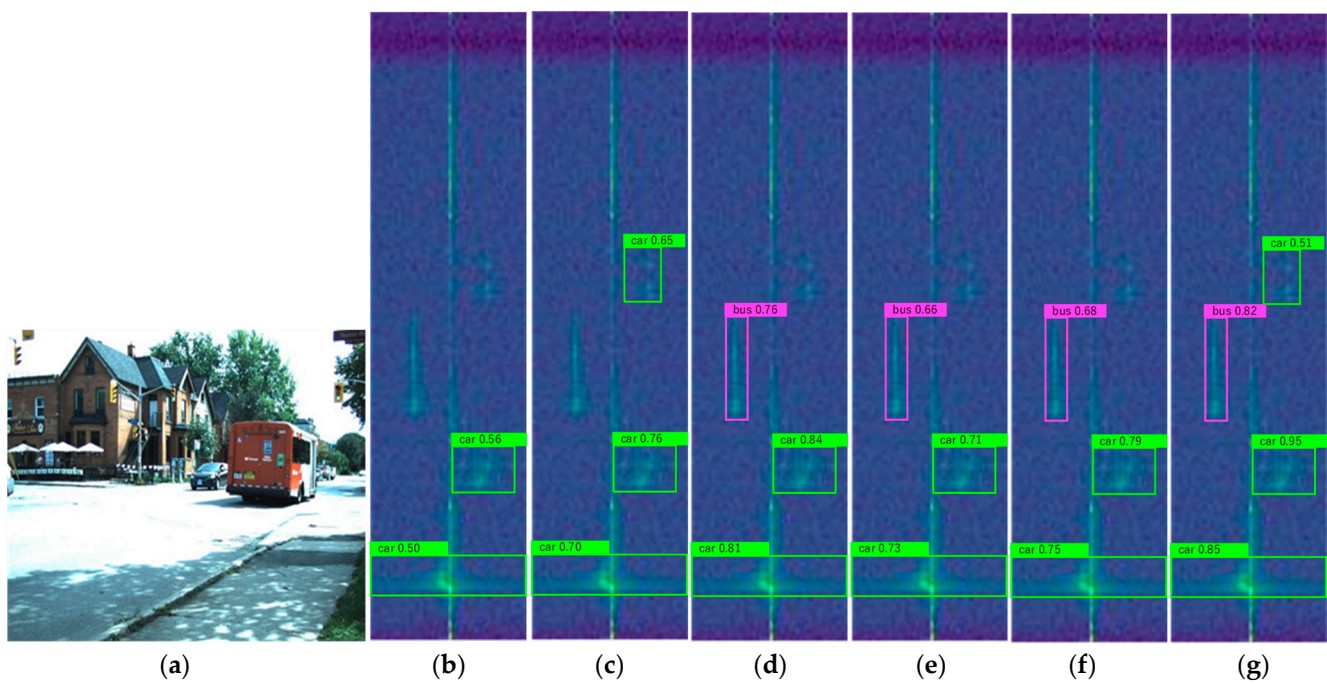

**Figure 7.** The comparison chart of the detection effect of different mainstream methods on the RADDet dataset. (**a**) The light-sensing image corresponding to the RD spectrum in the traffic scene, (**b**) the detection effect diagram of the Faster RCNN method, (**c**) the detection effect of YOLOv5, (**d**) the detection effect of YOLOv7, (**e**) the detection effect of YOLOv7-tiny, (**f**) the detection effect of YOLOv8, (**g**) the detection result of our proposed model.

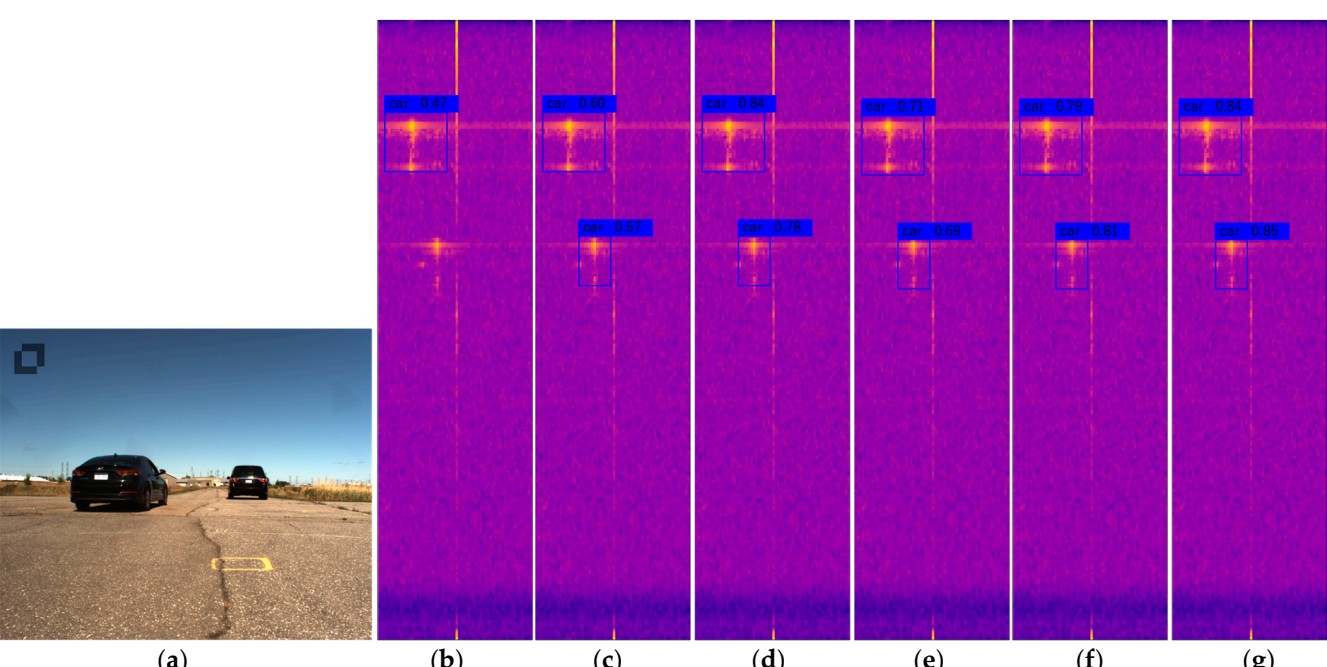

**Figure 8.** The comparison chart of the detection effects of different mainstream methods on the CARRADA dataset. (**a**) The light-sensing image corresponding to the RD spectrum in the traffic scene, (**b**) the detection effect diagram of the Faster RCNN method, (**c**) the detection effect of YOLOv5, (**d**) the detection effect of YOLOv7, (**e**) the detection effect of YOLOv7-tiny, (**f**) the detection effect of YOLOv8, (**g**) the detection result of our proposed model.

**Table 4.** Results of different models on RADDet and CARRADA datasets (the dataset division ratio is 9:1).

| Dataset | Model | mAP | IOU 0.3 P | R | mAP | IOU 0.5 P | R | Paras (M) | GFLOPs (G) | FPS (ms) | Training Time (s) | Testing Time (s) |
|---|---|---|---|---|---|---|---|---|---|---|---|---|
| RADDet | Faster RCNN | 56.47 | 52.17 | **56.92** | 49.55 | 47.78 | **51.77** | 41.3 | 60.1 | 26.9 | 607 | 68 |
| | YOLOv5 | 51.98 | 75.57 | 33.5 | 41.71 | 66.21 | 30.21 | 7.1 | 16.5 | 41.6 | 223 | 15 |
| | YOLOv7 | 69.76 | 85.69 | 45.68 | 58.1 | 75.67 | 46.62 | 37.2 | 105.2 | 34.4 | 457 | 32 |
| | YOLOv7tiny | 64.07 | 83.8 | 42.46 | 54.62 | 80.57 | 39.36 | 6.0 | 13.2 | 66.2 | 236 | 12 |
| | YOLOv8 | 67.94 | 94.26 | 29.76 | 57.13 | 91.56 | 29.13 | 25.9 | 79.1 | 41.1 | 294 | 31 |
| | RADDet | 38.42 | 78.2 | 29.77 | 22.87 | 60.41 | 20.55 | 7.8 | 5.0 | 13.5 | 621 | 54 |
| | DAROD | 65.56 | 82.31 | 47.78 | 46.57 | 68.23 | 38.74 | 3.4 | 6.8 | 39.5 | 286 | 20 |
| | Ours | **74.51** | 89.94 | 45.95 | **64.26** | 86.63 | 44.46 | 25.9 | 82 | 29.8 | 291 | 20 |
| CARRADA | Faster RCNN | 65.08 | 51.7 | **72.97** | 61.56 | 47.86 | **67.21** | 41.3 | 60.1 | 26.9 | 596 | 57 |
| | YOLOv5 | 49.08 | 76.69 | 31.11 | 40.16 | 65.68 | 29.76 | 7.1 | 16.5 | 41.6 | 199 | 14 |
| | YOLOv7 | 70.0 | 82.98 | 28.21 | 59.38 | 78.68 | 21.86 | 37.2 | 105.2 | 34.3 | 442 | 29 |
| | YOLOv7tiny | 64.37 | 86.32 | 26.66 | 55.46 | 77.88 | 34.36 | 6.0 | 13.2 | 66.2 | 230 | 20 |
| | YOLOv8 | 71.04 | 88.05 | 27.34 | 59.03 | 91.25 | 26.68 | 25.9 | 79.1 | 46.2 | 280 | 23 |
| | RADDet | 48.59 | 61.31 | 42.56 | 18.57 | 36.73 | 25.5 | 7.8 | 5.0 | 13.5 | 615 | 47 |
| | DAROD | 70.68 | 76.73 | 52.52 | 55.83 | 68.34 | 46.03 | 3.4 | 6.8 | 39.5 | 272 | 19 |
| | Ours | **75.62** | **94.54** | 38.29 | **62.54** | 90.55 | 33.47 | 25.9 | 82 | 30.6 | 253 | 18 |

**Table 5.** Results of different models on RADDet and CARRADA datasets (the dataset division ratio is 7:3).

| Dataset | Model | mAP | IOU 0.3 P | R | mAP | IOU 0.5 P | R | Paras (M) | GFLOPs (G) | FPS (ms) | Training Time (s) | Testing Time (s) |
|---|---|---|---|---|---|---|---|---|---|---|---|---|
| RADDet | Faster RCNN | 53.99 | 50.53 | **55.67** | 49.01 | 47.92 | **50.65** | 41.3 | 60.1 | 26.9 | 636 | 73 |
| | YOLOv5 | 52.02 | 74.86 | 33.35 | 42.27 | 63.67 | 31.51 | 7.1 | 16.5 | 41.6 | 194 | 16 |
| | YOLOv7 | 67.19 | 82.96 | 46.83 | 57.01 | 73.7 | 48.52 | 37.2 | 105.2 | 34.4 | 426 | 35 |
| | YOLOv7tiny | 62.18 | 83.6 | 40.13 | 54.36 | 79.1 | 38.96 | 6.0 | 13.2 | 66.2 | 264 | 29 |
| | YOLOv8 | 69.19 | 86.81 | 37.6 | 57.1 | 83.44 | 39.86 | 25.9 | 79.1 | 41.1 | 294 | 31 |
| | RADDet | 37.5 | 77.42 | 28.79 | 22.1 | 58.87 | 21.63 | 7.8 | 5.0 | 13.5 | 618 | 55 |
| | DAROD | 63.65 | 79.1 | 45.19 | 45.38 | 66.8 | 37.04 | 3.4 | 6.8 | 39.5 | 281 | 23 |
| | Ours | **73.6** | 89.17 | 44.21 | **63.68** | 83.09 | 46.52 | 25.9 | 82 | 29.8 | 283 | 24 |
| CARRADA | Faster RCNN | 66.21 | 46.98 | **69.15** | 61.44 | 45.38 | **63.86** | 41.3 | 60.1 | 26.9 | 587 | 62 |
| | YOLOv5 | 50.24 | 75.35 | 34.07 | 41.36 | 65.99 | 28.81 | 7.1 | 16.5 | 41.6 | 179 | 15 |
| | YOLOv7 | 67.8 | 81.61 | 29.04 | 56.75 | 77.92 | 20.11 | 37.2 | 105.2 | 34.3 | 399 | 32 |
| | YOLOv7tiny | 62.89 | 81.57 | 24.16 | 54.71 | 74.61 | 30.1 | 6.0 | 13.2 | 66.2 | 190 | 18 |
| | YOLOv8 | 66.7 | 75.8 | 31.37 | 55.71 | 86.08 | 27.53 | 25.9 | 79.1 | 46.2 | 282 | 25 |
| | RADDet | 46.72 | 60.27 | 40.3 | 18.46 | 35.19 | 24.44 | 7.8 | 5.0 | 13.5 | 609 | 45 |
| | DAROD | 66.56 | 74.31 | 50.68 | 51.8 | 65.34 | 46.62 | 3.4 | 6.8 | 39.5 | 257 | 19 |
| | Ours | **71.81** | 92.9 | 37.54 | 60.37 | **88.19** | 32.17 | 25.9 | 82 | 30.6 | 260 | 22 |

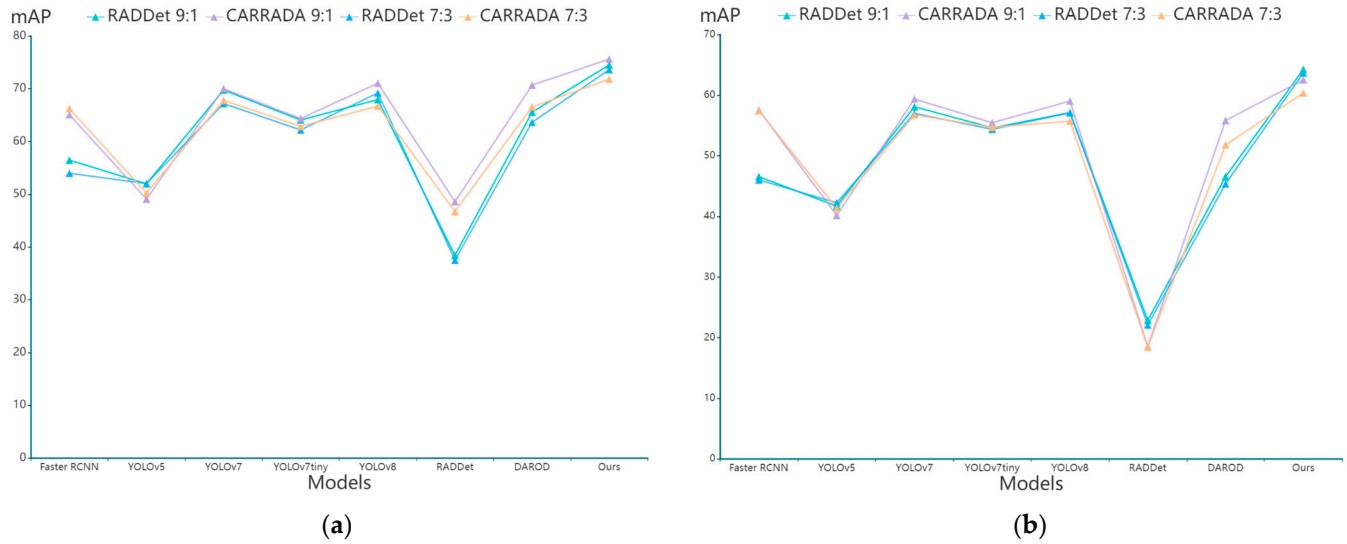

**Figure 9.** Two graphs comparing the performance of each model. (**a**) Indicates the performance of each model when the IOU is 0.3, and the RADDet and CARRADA datasets are divided in ratios of 9:1 and 7:3, respectively. (**b**) Indicates the performance of each model when the IOU is 0.5, and the RADDet and CARRADA datasets are divided in ratios of 9:1 and 7:3, respectively.

Overall, our model achieves the best mAP on both datasets and far exceeds the mAP of other modeling methods. Although it may not be the best in terms of detection precision and recall compared to other methods, the precision and recall of our model are

not far behind the best results of other models. Moreover, mAP can measure the overall performance of the entire model detection. It does not mean that the model is good if a single accuracy or recall rate is good.

Specifically, on the RADDet dataset, our model achieved the best results when the IOU was 0.3 and 0.5, but the detection accuracy was not as good as that of the YOLOv8 model. But the gap is not that big, and the recall of our model is much higher than YOLOv8. There are gains and losses. Our model's FPS (frames per second) is lower than YOLOv8, and GFLOPS (floating-point operations) is higher. Compared with YOLOv7, our model index is overall higher than YOLOv7. Our model has the best overall performance, and YOLOv7's overall performance ranks second, but its FPS performance is better, which may be more complicated than our model, so the calculation speed and inference speed are slightly reduced. For the time complexity, we comprehensively consider the training time of each model for one cycle. When the ratio of the training set to the test set is 9:1, on the two datasets, the YOLOv5 model requires the least training time and the time complexity is low, followed by the lightweight YOLOv7-tiny model. Because our model is improved based on the YOLOv8 structure, the model is more complex, so the required training time is also slightly longer, but it is less complex than the YOLOv8 model. In terms of model reasoning, YOLOv7-tiny is the fastest on the RADDet dataset, followed by YOLOv5; on the CARRADA dataset, YOLOv5 is the fastest, followed by our model. When the ratio of the training set to the test set in the dataset is 7:3, the fastest training time is still YOLOv5, followed by YOLOv7-tiny, and our model ranks third. In terms of reasoning, YOLOv5 reasoning is the fastest on the RADDet dataset, followed by DAROD, and our model ranks third. The fastest inference on the CARRADA dataset is still YOLOv5, followed by YOLOv7-tiny, and our model ranks third. Overall, although the time complexity of our model is not the lowest, it is slightly higher than both YOLOv5 and YOLOv7. In terms of overall performance, our model detection effect is the best.

On the CARRADA dataset, the overall performance of our model is still the best. Although the performance of Faster RCNN is also good, the two models can compete with each other, but overall, the mAP, accuracy, model parameters, and FPS of our model are better than Faster RCNN, and it can detect the target to the greatest extent. Faster RCNN has higher recall, which leads to more false positives but fewer missed objects.

In order to improve the reliability of the experimental data, we also counted the movement speed of each category in the two datasets, and then analyzed the adaptability of our model to targets with different movement speeds. Figure 10a,b show the velocity distribution statistics of all targets in the RADDet and CARRADA datasets, respectively. We used the Doppler dimension label data for analysis and statistics and calculated the raw velocity of the object motion based on the Doppler information given by each target label. From Figure 10, we can know that the speed of the moving target in the traffic scene is roughly between $-13.5$ m/s and $13.5$ m/s.

According to the speed information provided in Figure 10, combined with the detection results of each model in Figure 10, it can be seen that in the traffic road scene, for moving objects or stationary objects with different speeds, our proposed model outperforms other models in terms of overall performance.

For the two models of RADDet and DAROD, their mAP, accuracy, and recall are not outstanding, but the advantage is that the model parameters are small and the GFLOPs are very small, conducive to model deployment. However, our model and other models still need to be improved in terms of deployment.

In conclusion, compared with other models, our model achieves better mAP and detection accuracy on two datasets, medium recall rate, and medium model parameters, GFLOPS and FPS. This shows that our model is more accurate in the target detection task. However, due to the attributes of RD data, it is difficult for us to detect all targets, and there are still missed and false detections.

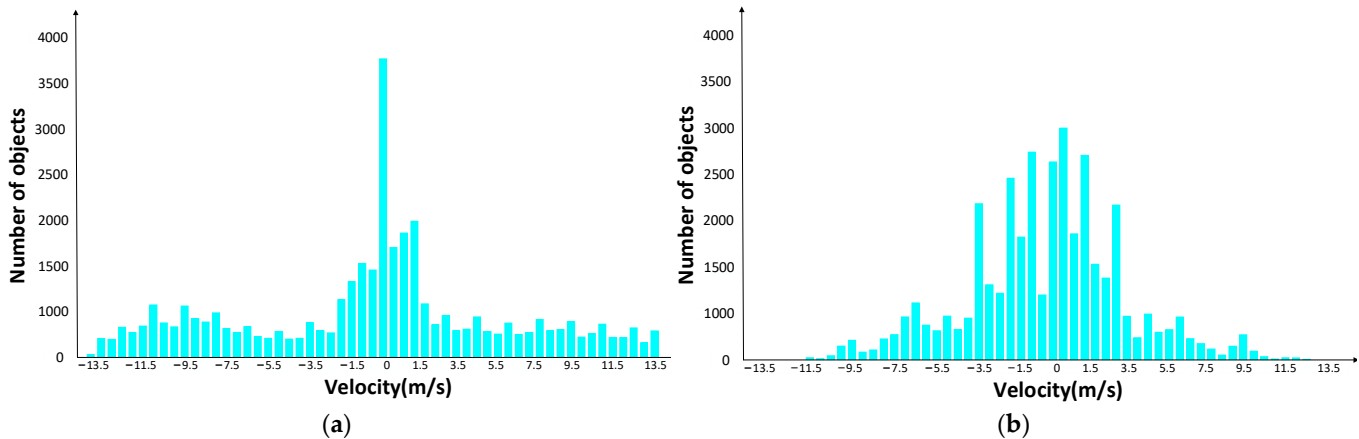

**Figure 10.** Statistical distribution maps of target speed in the datasets. (**a**) Statistical distribution map of target speed in RADDET dataset; (**b**) statistical distribution map of target speed in CAR-RADA dataset.

### 4.5. Ablation Experiments

In this subsection, in order to know exactly the contribution of each module in our model, we will briefly introduce the performance exhibited by different network modules. Tables 6 and 7 list the results obtained by different modules on the RADDet and CAR-RADA datasets. Figures 11 and 12 show the prediction results of different modules on the RADDet and CARRADA datasets. In the figures, "YOLOv8" represents the original model architecture, "ConvLSTM" represents the network that only introduces the ConvLSTM module in the YOLOv8 backbone network, "ECA" represents the network that introduces the ECA attention mechanism in the three feature fusion layers output by the backbone network, and "Ours" represents the model structure we proposed. Figure 13 shows the line chart of the mAP performance of different models, which illustrates the effectiveness of our proposed model.

**Table 6.** Improved experiments based on YOLOv8 (the dataset division ratio is 9:1).

| Dataset | Model | mAP | IOU 0.3 P | R | mAP | IOU 0.5 P | R | Paras (M) | GFLOPs (G) | FPS (ms) | Training Time (s) | Testing Time (s) |
|---|---|---|---|---|---|---|---|---|---|---|---|---|
| RADDet | YOLOv8 | 67.94 | 94.26 | 29.76 | 57.13 | 91.56 | 29.13 | **25.86** | 79.08 | 41.12 | 294 | 31 |
| | ConvLSTM | 70.32 | 92.32 | 44.5 | 60.59 | **93.86** | 36.36 | 25.92 | 82.01 | 29.99 | **287** | **18** |
| | ECA | 70.33 | **95.67** | 41.12 | 60.6 | 92.39 | 39.99 | 25.86 | 79.09 | 39.56 | 300 | 19 |
| | Ours | **74.51** | 89.94 | **45.95** | **64.26** | 86.63 | **44.46** | 25.92 | 82.01 | 29.81 | 291 | 20 |
| CARRADA | YOLOv8 | 71.04 | 88.05 | 27.34 | 59.03 | **91.25** | 26.68 | **25.86** | 79.08 | 46.17 | 280 | 23 |
| | ConvLSTM | 73.93 | 93.35 | 32.57 | 60.06 | 88 | 24.53 | 25.92 | 82.01 | 29.96 | 254 | 18 |
| | ECA | 74.05 | 93.08 | 32.29 | 60.71 | 89.75 | 31.04 | 25.86 | 79.09 | 41.68 | 260 | 18 |
| | Ours | **75.62** | **94.54** | **38.29** | **62.54** | 90.55 | **33.47** | 25.92 | 82.01 | 30.62 | **253** | **18** |

**Table 7.** Improved experiments based on YOLOv8 (the dataset division ratio is 7:3).

| Dataset | Model | mAP | IOU 0.3 P | R | mAP | IOU 0.5 P | R | Paras (M) | GFLOPs (G) | FPS (ms) | Training Time (s) | Testing Time (s) |
|---|---|---|---|---|---|---|---|---|---|---|---|---|
| RADDet | YOLOv8 | 69.19 | 86.81 | 37.6 | 57.1 | 83.44 | 39.86 | **25.86** | 79.08 | **41.12** | 294 | 31 |
| | ConvLSTM | 69.57 | 90.8 | 43.03 | 56.29 | 83.65 | 40.24 | 25.92 | 82.01 | 29.99 | 285 | 26 |
| | ECA | 68.34 | **92.08** | **46.59** | 55.06 | **86.06** | 43.58 | 25.86 | 79.09 | 39.56 | 287 | 29 |
| | Ours | **73.6** | 89.17 | 44.21 | **63.68** | 83.09 | **46.52** | 25.92 | 82.01 | 29.81 | **283** | **24** |
| CARRADA | YOLOv8 | 66.7 | 75.8 | 31.37 | 55.71 | 86.08 | 27.53 | **25.86** | 79.08 | 46.17 | 282 | 25 |
| | ConvLSTM | 69.79 | 89.82 | 30.91 | 56.13 | 86.65 | 23.19 | 25.92 | 82.01 | 29.96 | **247** | 24 |
| | ECA | 71.16 | 90.66 | 33.29 | 58.62 | **91.91** | 30.8 | 25.86 | 79.09 | 41.68 | 256 | **20** |
| | Ours | **71.81** | **92.9** | **37.54** | **60.37** | 88.19 | **32.17** | 25.92 | 82.01 | 30.62 | 260 | 22 |

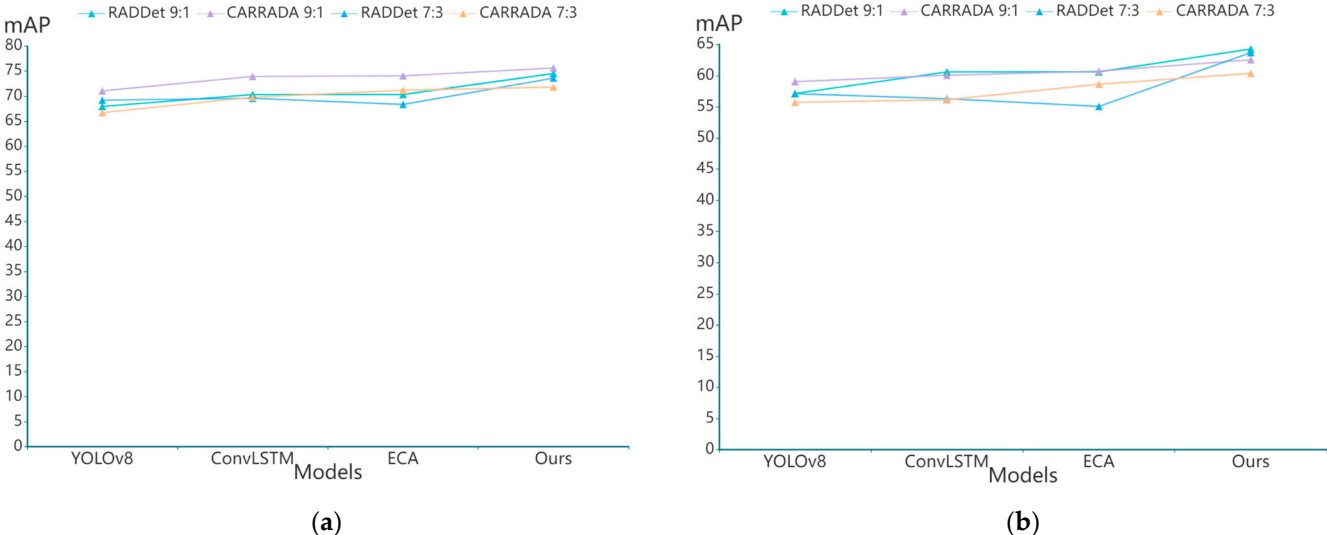

(**a**)                                                      (**b**)

**Figure 11.** Two graphs comparing the performance of each model. (**a**) The performance of the comparison model when the IOU is 0.3, and the RADDet and CARRADA datasets are divided in ratios of 9:1 and 7:3, respectively. (**b**) The performance of the comparison model when the IOU is 0.5, and the RADDet and CARRADA datasets are divided in ratios of 9:1 and 7:3, respectively.

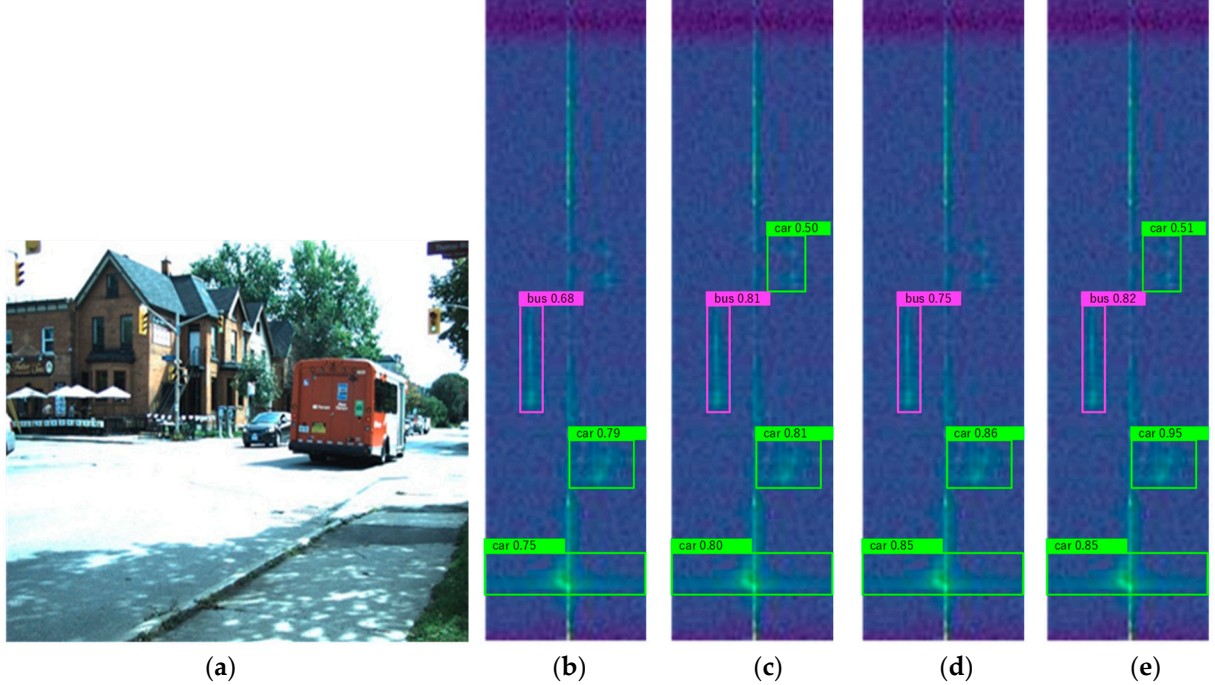

(**a**)                     (**b**)                     (**c**)                     (**d**)                     (**e**)

**Figure 12.** The prediction results of different models on the RADDet dataset. (**a**) The light-sensing image corresponding to the RD spectrum in the traffic scene, (**b**) the prediction graph of the YOLOv8 model, (**c**) the prediction graph of the LSTM model, (**d**) the prediction graph of the ECA model, (**e**) the prediction graph of our proposed model.

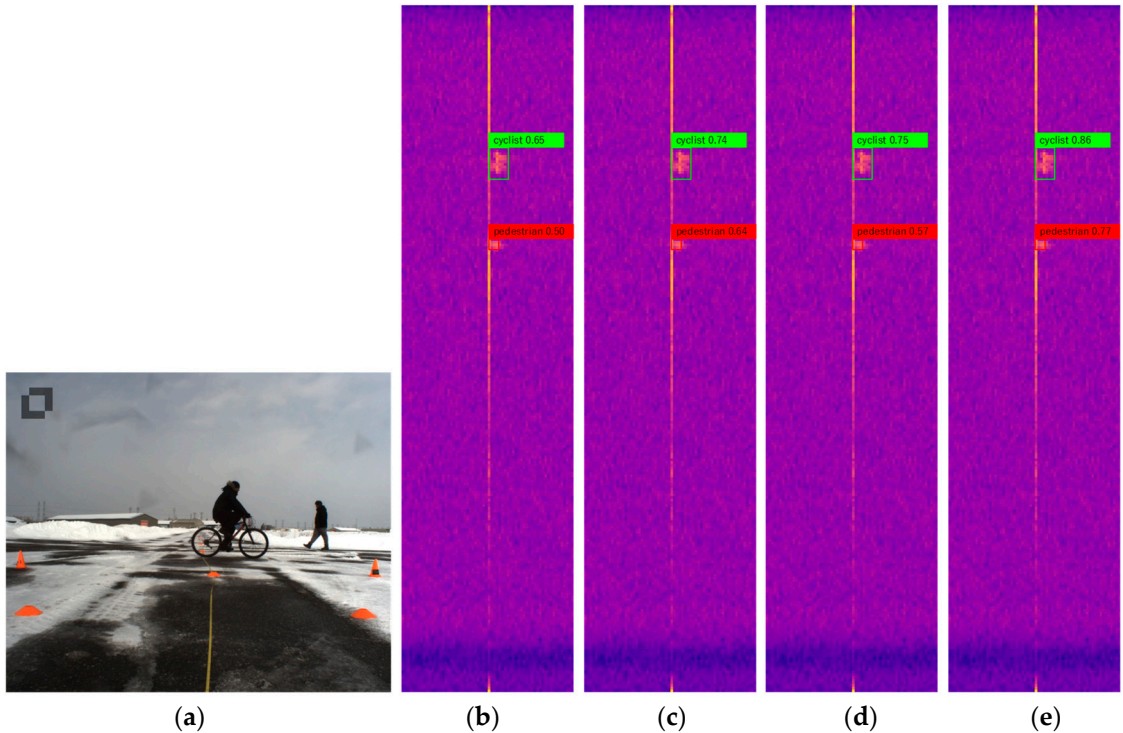

**Figure 13.** The prediction results of different models on the CARRADA dataset. (**a**) The light-sensing image corresponding to the RD spectrum in the traffic scene, (**b**) the prediction graph of the YOLOv8 model, (**c**) the prediction graph of the LSTM model, (**d**) the prediction graph of the ECA model, (**e**) the prediction graph of our proposed model.

Our model is mainly based on the improvement of YOLOv8, and ConvLSTM and ECA are the main modules that play a role. Specifically, it can be seen from Tables 6 and 7 that the improvement of YOLOv8 by ConvLSTM and ECA is roughly similar, but ECA can be improved with a small change in the parameters of YOLOv8 and FPS, which improves the model's attention to key features. ConvLSTM slightly increases the number of parameters of the original model and reduces the FPS, because ConvLSTM introduces additional parameters when it plays a role in memory of time series data. It can be seen from Figure 12 that YOLOv8 has missed detection. When the same ECA works alone, it still does not detect the missed detection target, and only improves the accuracy of the detected target. When ConvLSTM works alone, the missed detection target can be detected, and the detection confidence is generally improved, which shows that ConvLSTM has exerted the ability of time series feature extraction. Combining Figures 11–13, it can be seen that when only ECA and ConvLSTM work alone, the target detection accuracy in the two datasets is not high, even lower than the YOLOv8 model, which reduces the detection accuracy of the original model. Our model combines the advantages of ECA and ConvLSTM. It not only utilizes timing to reduce the missed detection rate, but also improves the accuracy of object detection, enabling the model to maximize its performance. When changing the distribution ratio of the training set and the verification set in the dataset, from the original 9:1 to 7:3, that is, when the training data ratio decreases, the changes in each model are the same, which fully shows that our model is very reliable.

In order to improve the generality of the model, compared with Figures 12 and 13, Figures 14 and 15 show the target detection of the radar device placed on another traffic road in the RADDet dataset and the CARRADA dataset, respectively. From the detection comparison results in the four pictures, it can be seen that for target detection in different scenarios, the first three models will have the disadvantages of missed detection or low accuracy. Our model can always combine the timing of ConvLSTM and the accuracy of

ECA to achieve the best detection effect and the highest detection accuracy, and greatly improve the detection accuracy on the basis of the YOLOv8 model.

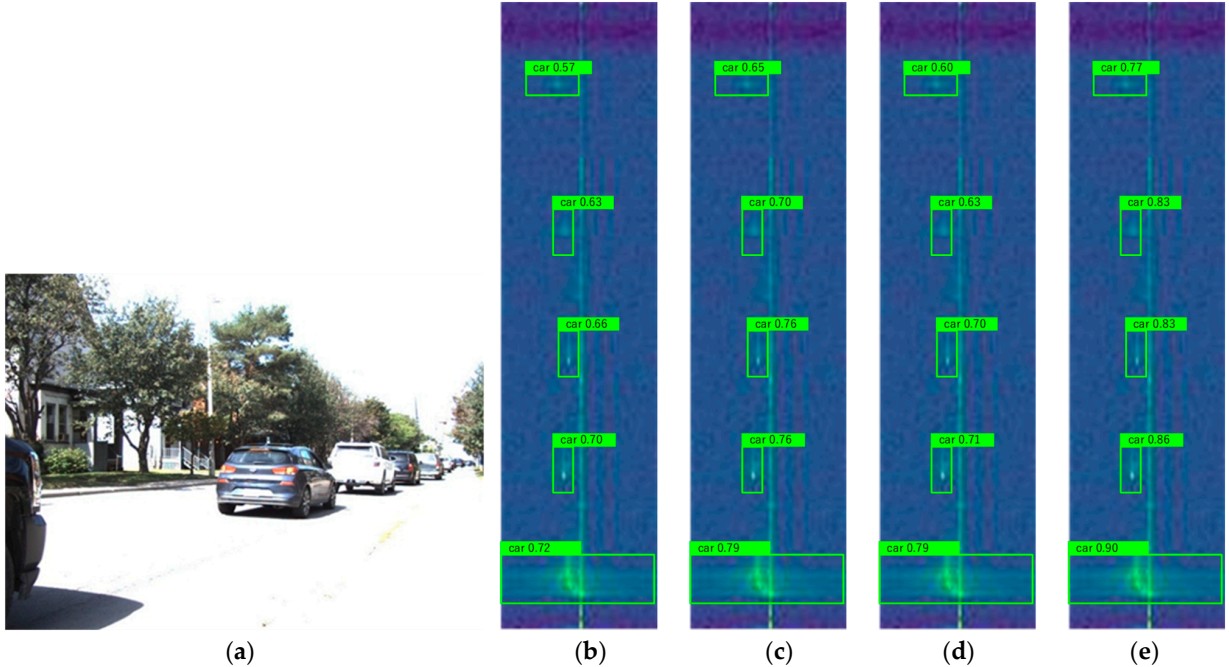

**Figure 14.** The prediction results of different models on the RADDet dataset. (**a**) The light-sensing image corresponding to the RD spectrum in the traffic scene, (**b**) the prediction graph of the YOLOv8 model, (**c**) the prediction graph of the LSTM model, (**d**) the prediction graph of the ECA model, (**e**) the prediction graph of our proposed model.

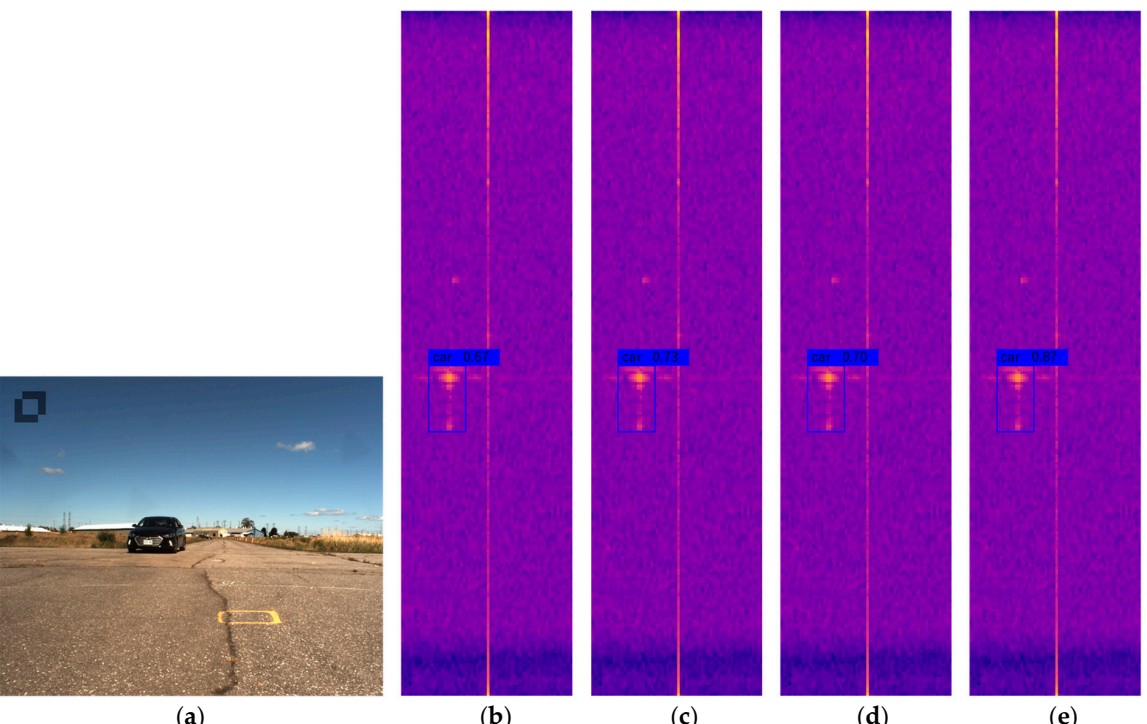

**Figure 15.** The prediction results of different models on the CARRADA dataset. (**a**) The light-sensing image corresponding to the RD spectrum in the traffic scene, (**b**) the prediction graph of the YOLOv8 model, (**c**) the prediction graph of the LSTM model, (**d**) the prediction graph of the ECA model, (**e**) the prediction graph of our proposed model.

In short, ConvLSTM learns timing information during feature extraction, and three ECA attention modules further improve the network feature mining ability. Although the two parts of the modules can improve the network performance when they play their respective roles, the improvement to the network is not large. In most cases, only when these two modules work together can the network perform stably and efficiently. Not only that, the training speed and inference speed of the network are also greatly improved in the case of synergy, which effectively proves the superiority of the proposed method. In summary, ablation experiments show that our proposed network model is beneficial for object detection in radar RD spectra of traffic scenes.

## 5. Conclusions

In this paper, a deep-learning-based object detection method based on radar range–Doppler spectroscopy is proposed. According to the characteristics of the radar range–Doppler spectrum, we have made some improvements on the basis of the YOLOv8 model. The detection performance of the model is mainly improved by the modification of the backbone network and the optimization of other parts. Compared with other currently known RD map object detection models, our model fully utilizes the temporal information of the traffic scene dataset, and improves the ability of the model to focus on key objects through the attention mechanism. Since there are few references to the radar range–Doppler spectrum dataset, we used two datasets from foreign traffic scenarios, namely RADDet and CARRADA datasets. We processed the raw radar data provided in the dataset and turned them into a visualized range–Doppler spectrum, on which our experiments were based. We compared the current mainstream target detection methods, namely Faster RCNN, YOLOv5, YOLOv7, YOLOv7-tiny, YOLOv8, RADDet, and DAROD, with our proposed model. In order to observe the effectiveness of our model, we split the dataset into a training set and a testing set with ratios of 9:1 and 7:3, respectively. The experimental results show that the improved model not only improves the detection accuracy, but also the model parameters and complexity are at a reasonable level. The results of inference on the range–Doppler spectra in the test set show that our proposed model has better detection performance and robustness compared to other methods with a low detection rate and missed detection stickiness. The experiments prove the effectiveness and feasibility of our proposed method, conducive to the smooth progress of object detection in actual traffic scenes. However, our proposed method increases the number of parameters and model complexity, making it difficult to deploy, and we hope to improve it in future work. At the same time, we will continue to pay attention to the characteristics of the target in the radar range–Doppler spectrum and propose a more targeted optimization strategy, collect data on the basis of perfect basic radar equipment, further expand the dataset, and involve more target types.

**Author Contributions:** Conceptualization, F.J. and J.T.; investigation, J.T. and X.L.; methodology, F.J. and J.T.; writing—original draft and formal analysis, F.J. and J.T.; writing—review and editing, J.Q. and X.L. All authors have read and agreed to the published version of the manuscript.

**Funding:** This work was supported by the National Natural Science Foundation of China (grants 62001064).

**Data Availability Statement:** Not applicable.

**Acknowledgments:** The authors would like to thank the reviewers and editor for their valuable comments and suggestions.

**Conflicts of Interest:** The authors declare no conflict of interest.

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
