# Peer review of "Radar Timing Range–Doppler Spectral Target Detection Based on Attention ConvLSTM in Traffic Scenes"

_remotesensing, doi:10.3390/rs15174150_

Round 1

Reviewer 1 Report

In this work, we propose a deep learning-based object detection method with respect to radar Range-Doppler spectrum in traffic scenarios. However, some weaknesses should be addressed, especially the introduction and experiment.

1) In INTRODUCTION part, the introduction of background is too simple. Many of the latest work has not been introduced, which is not enough to fully explain the significance of this study. In addition, the technology of object detection in other image fields should also be introduced. Therefore, the authors are suggested to add some literatures. e. g.,

[1] Super-Resolution Mapping Based on Spatial-Spectral Correlation for Spectral Imagery [J]. IEEE Transactions on Geoscience and Remote Sensing, 2021, 59(3): 2256-2268.

[2] Target-Constrained Interference-Minimized Band Selection for Hyperspectral Target Detection, IEEE Transactions on Geoscience and Remote Sensing, 2021, 59(7): 6044-6064.

2) In the method section, what are the innovations of the change detection methods currently used by the authors? It is recommended that the author focus on describing which module of the designed deep learning network is innovative? Why is this design?

3) The experimental part needs to be improved. Can you add more ablation experiments? Can you compare it with some similar latest methods? In addition, the authors only conduct experiments on two data, and the versatility of the method's performance cannot be well demonstrated. It is recommended that the authors add more dataset experiments.

 4) What is the computational complexity of the proposed method? Suggest providing relevant calculation time.

Reviewer 2 Report

This manuscript introduces a deep learning based radar RD spectrum target detection method. The proposed technique was able to improve accuracy in traffic scenarios through the ConvLSTM network and ECA. However, there are some difficulties in understanding the manuscript due to lack of explanation and analysis, so I suggested some parts that need to be added.

- It seems that re-implementation of this simulation is not easy due to a lack of detailed explanation of the experiment. It is necessary to organize the experimental environment, especially the size of the data, in a table.

- It is necessary to refine abstracts and conclusions by reflecting factors such as research motivation, utility of proposal techniques, and simulation results.

- Various traffic speeds need to be considered to identify the characteristics of the proposed technique.

- Analytical content should be added on what effects of the proposed techniques can improve accuracy.

Some expressions such as ‘Xt, Ht-1, Ct-1’ on 8 pages need to be organized into equations.

Round 2

Reviewer 1 Report

Thank you for the authors‘ reply. I don't have any other questions.

Reviewer 2 Report

The paper has been revised well. I recommend publishing this paper.